

**Formation, radiative forcing, and climatic effects of severe regional haze**
Yun Lin[1,2*], Yuan Wang[3*], Bowen Pan[1,4], Jiaxi Hu[1,5], Song Guo[6], Misti Levy Zamora[1,7],
Pengfei Tian[1,8], Qiong Su[9], Yuemeng Ji[1,10], Jiayun Zhao[11], Mario Gomez-Hernandez[11], Min
Hu[6], Renyi Zhang[1,11*]
[1]Department of Atmospheric Sciences, Texas A&M University, College Station, TX 77843,
USA
[2]Joint Institute for Regional Earth System Science and Engineering (JIFRESSE), University
of California at Los Angeles, Los Angeles, CA 90064
[3]Division of Geological and Planetary Sciences, California Institute of Technology,
Pasadena, CA 91125, USA
[4]Department of Atmospheric Science, Colorado State University, Fort Collins, CO, 80521,
USA
[5]Cooperative Institute for Mesoscale Meteorological Studies, NOAA/OAR National Severe
Storms Laboratory, Norman, OK, USA
[6]State Key Joint Laboratory of Environmental Simulation and Pollution Control, College of
Environmental Sciences and Engineering, Peking University, Beijing, 10087, P.R. China
[7]Department of Environmental Health and Engineering, Johns Hopkins Bloomberg School of
Public Health, 615 N. Wolfe St., Baltimore, MD 21205, USA
[8]Key Laboratory for Semi-Arid Climate Change of the Ministry of Education, College of
Atmospheric Sciences, Lanzhou University, Lanzhou 730000, P. R. China
[9]Water Management & Hydrological Science, Texas A&M University, College Station, TX
77843, USA
[10]Guangzhou Key Laboratory of Environmental Catalysis and Pollution Control, School of
Environmental Science and Engineering, Institute of Environmental Health and Pollution
Control, Guangdong University of Technology, Guangzhou 510006, China
[11]Department of Chemistry, Texas A&M University, College Station, TX 77843, USA
*Correspondence: yunlin@ucla.edu; Yuan.Wang@caltech.edu; renyi-zhang@tamu.edu



**Abstract.** Severe regional haze events, which are characterized by exceedingly high levels of
fine particulate matter (PM), occur frequently in many developing countries (such as China
and India), with profound implications for human health, weather, and climate. The occurrence
of the haze extremes involves a complex interplay between primary emissions, secondary
formation, and conducive meteorological conditions, and the relative contributions of the
various processes remains unclear. Here we investigated severe regional haze episodes in 2013
over the Northern China Plain (NCP), by evaluating the PM production and the interactions
between elevated PM and the planetary boundary layer (PBL). Analysis of the ground-based
measurements and satellite observations of PM properties shows nearly synchronized temporal
PM variations among the three megacities (Beijing, Baoding, and Shijiazhuang) in this region
and a coincidence of the aerosol optical depth (AOD) hotspots with the three megacities during
the polluted period. During the clean-to-hazy transition, the measured oxygenated organic
aerosol concentration ([OOA]) well correlates with the odd-oxygen concentration ($[O_x] = [O_3]$
$+ [NO_2]$), and the mean $[OOA]/[O_x]$ ratio in Beijing is much larger than those in other
megacities (such as Mexico City and Houston), indicating highly efficient photochemical
activity. Simulations using the Weather Research and Forecasting (WRF) model coupled with
an explicit aerosol radiative module reveal that strong aerosol-PBL interaction during the
polluted period results in a suppressed and stabilized PBL and elevated humidity, triggering a
positive feedback to amplify the haze severity at the ground level. Model sensitivity study
illustrates the importance of black carbon (BC) in the haze-PBL interaction and the aerosol
regional climatic effect, contributing to more than 30% of the PBL collapse and about half of
the positive radiative forcing on the top of the atmosphere. Overall, severe regional haze
exhibits strong negative radiative forcing (cooling) of -63 to -88 W m$^{-2}$ at the surface and strong
positive radiative forcing (warming) of 57 to 82 W m$^{-2}$ in the atmosphere, with a slightly
negative net radiative forcing of about -6 W m$^{-2}$ on the top of the atmosphere. Our work



establishes a synthetic view for the dominant regional features during severe haze events,
unraveling rapid *in-situ* PM production and inefficient transport, both of which are amplified
by atmospheric stagnation. On the other hand, regional transport sufficiently disperses gaseous
aerosol precursors (e.g., sulfur dioxide, nitrogen oxides, volatile organic compounds, and
ammonia) during the clean period, which subsequently result in rapid *in-situ* PM production
via photochemistry during the transition period and via multiphase chemistry during the
polluted period. Our findings highlight the co-benefits for reduction in BC emissions, which
not only improve local and regional air quality by minimizing air stagnation but also mitigate
the global warming by alleviating the positive direct radiative forcing.





## 1. Introduction


Rapid economic growth and urbanization have caused frequent severe regional haze
events associated with heavy pollution of particulate matter (PM) in many developing countries,
including China and India (Bouarar et al., 2017; Molina, 2021). The severe haze events induce
great degradation in visibility and air quality, with profound societal implications (An et al.,
2019). For example, exposure to elevated levels of fine PM leads to adverse health effects,
ranging from aggravated allergies to the development of chronic diseases, to premature death
(Pope and Dockery, 2015; Wu et al., 2019; Rychlik et al., 2019; Johnson et al., 2021; Zhang et
al., 2021). Also, elevated levels of fine aerosols result in pronounced modifications to clouds,
precipitation, and lightning, impacting regional/global weather and climate (Zhang et al., 2007;
Yuan et al., 2008; Qian et al., 2009; Wang et al., 2011; Wang et al., 2014; Wu et al., 2016).
Specifically, by absorbing/scattering solar radiation, aerosols impact the atmospheric stability
and the energy budget of Earth, via the aerosol-radiation interaction (ARI). By serving as cloud
condensation nuclei (CCN) and ice nucleating particles (INPs), aerosols influence the macro-
and microphysical properties of clouds, via the aerosol–cloud interaction (ACI). Currently, the
radiative forcing associated with ARI and ACI represents the largest uncertainty in the
projection of future climate by anthropogenic activities (IPCC, 2013).
PM is either emitted directly into the atmosphere (primary) or produced in air via gas-
to-particle conversion (secondary) (Zhang et al., 2015a). In addition, primary and secondary
PM undergo chemical and physical transformations and are subjected to cloud processing and
removal from air (Zhang et al., 2015a). Direct emissions of primary gases and PM and highly
efficient secondary PM formation represent the primary processes leading to severe haze (Guo
et al., 2014; Sun et al., 2014; Wang et al., 2016a; Peng et al., 2021). In addition, conducive
weather conditions for pollutant accumulation, such as regional control by high-pressure,
suppressed local circulations, and weakened large-scale circulation, correspond to the external


causes for severe haze formation (Liu et al., 2013; Wang et al., 2014d;  Cai et al., 2017; Li et
al., 2019).

The key constituents of fine PM include secondary inorganic (including sulfate, nitrate,

and ammonium) aerosol (SIA) and secondary organic aerosol (SOA), with the corresponding
gaseous precursors of sulfur dioxide ($SO_2$), nitrogen oxides ($NO_x = NO + NO_2$), ammonia
($NH_3$), and volatile organic compounds (VOCs). The photochemistry represents one of the
mechanisms leading to SIA and SOA accumulation during the early stage of haze evolution
(Guo et al., 2014; Zhang et al., 2015a;   Wang  et al., 2016; Zhang et al., 2020). Field
measurements have shown that remarkably nucleation and growth of nanoparticles are
primarily driven by photochemical activity, which is characterized by elevated ozone levels
and efficient photolysis rate coefficients under clean daytime conditions (Zhang et al., 2015b;
Guo et al., 2020). During haze evolution, the photochemical activity is typically reduced, as
evident by low levels of ozone and reduced photolysis rates (Peng et al., 2021). On the other
hand, there are increasing air stagnation and relative humidity (RH), when explosive secondary
aerosol formation occurs (Peng et al., 2021). The latter has been attributed the occurrence of
multiphase chemistry, which largely drives the formation of SIA and SOA during the polluted
period (Peng et al., 2021). Currently, the relative contributions of primary emissions, secondary
production, and regional transport to severe haze formation remain uncertain (Li et al., 2015;
Zhang et al., 2015b; Peng et al., 2021). Moreover, the efficiency of photochemical PM
production during regional haze events in NCP and its distinction among various megacities
worldwide remain to be quantified (Molina, 2021).

While the importance of regional haze on climate has been recognized (Ramanathan et

al., 2007; Wang et al., 2009; Wang et al., 2015a), there still lacks quantification for the aerosol
radiative forcing and the climatic effects for severe regional haze events. Estimation of the
aerosol radiative forcing during severe haze events exhibits a large variation (Li et al., 2007;



Xia et al., 2007; Wang et al., 2009; Che et al., 2014). In addition, the interactions between
aerosols and planetary boundary layer (PBL) via the aerosol radiative effects likely increase
the haze severity (Wang et al., 2015a; Wang et al., 2016b; Zhang et al., 2018). Meteorological
conditions within the PBL, including the atmospheric stability and RH, are altered by the
aerosol-PBL interaction to induce a positive feedback to PM accumulation near the ground
level (Tang et al., 2016a; Tie et al., 2017; Wu et al., 2020). However, the aerosol-PBL
interactions and their feedbacks to atmospheric thermodynamics and dynamics under
extremely hazy conditions remain to be quantified (Li et al., 2017).

Previous studies have documented the role of black carbon (BC) in the aerosol-PBL

interactions and the aerosol regional climate effects (Menon et al., 2002; Bond et al., 2013;
Wang et al., 2013; Ding et al., 2016). In addition, the BC aging process markedly enhances BC
absorption by modifying the particle physiochemical and optical properties (Zhang et al., 2008;
Khalizov et al., 2013; He et al., 2015; Guo et al., 2016; Peng et al., 2016; Peng et al., 2017).
For example, an experimental/field study showed that the mass absorption cross section (MAC)
of BC is enhanced by 2.4 times in a short time because of BC aging under polluted urban
conditions (Peng et al., 2016), reconciling previous variable results on the coating-enhanced
absorption for BC (Gustafsson and Ramanathan, 2016). Apparently, the enhancement of the
BC absorption causes additional aerosol radiative forcing (Peng et al., 2016) and suppression
on PBL development (Wang et al., 2017). Currently, limited modeling studies have assessed
the radiative effect of BC aging associated with severe regional haze (Wang et al., 2013; He et
al., 2015; Gustafsson and Ramanathan, 2016).

To better understand the formation and evolution of severe regional haze as well as

their regional and climate effects, we investigated severe haze episodes occurring in 2013 over
the Northern China Plain (NCP). The NCP region, which encompasses the megacities of
Beijing and Tianjin, and some portion of the provinces of Heibei, Shandong, and Henan,



represents the most polluted area in China (An et al., 2019). Satellite observations and field
measurements of PM properties were evaluated, and numerical simulations were performed to
elucidate the interactions between severe haze and PBL using Weather Research and Forecast
(WRF) model coupled with an explicit aerosol radiative module (Fan et al., 2008; Wang et al.,
2014c). By conducting model sensitivity simulations, we elucidated the impacts of BC aging
on the haze-PBL interactions and its contribution to the net aerosol radiative forcing during
severe haze periods.
**2. Methodology**
The NCP represents a key economic zone in China, as reflected by its gross domestic
product (GDP), energy consumption, and vehicular fleets (An et al., 2019). The region has
undergone fast industrialization and urbanization over the past four decades. For example, NCP
is one of the most densely populated regions in the world and contributes to over 1/10 of the
GDP in China. The consumption of coal and crude oil in NCP was 363 and 72 million tons,
respectively, to 1,348 million tons in 1998 and increased to 140 million tons of standard coal
equivalent in 2010, respectively. In particular, anthropogenic activities result in industrial,
traffic, residential, and agricultural emissions, representing the major sources for PM
precursors, including $SO_2$, $NO_x$, VOCs, and $NH_3$ (An et al., 2019; Peng et al., 2021).
Surrounded by the Taihang Mountains to the west and Yanshan Mountains to the north,
respectively, the NCP region is prone to develop air stagnation under conducive meteorological
conditions, inhibiting vertical and horizontal dispersion of air pollutants (An et al., 2019; Peng
et al., 2021).
**2.1. The Data Sources**
The satellite-retrieved aerosol optical depth (AOD) was derived by combining the
Moderate Resolution Imaging Spectroradiometer (MODIS) measurements of Aqua and Terra
using the equal-weighted mean method to increase the spatial coverage (Levy et al., 2009). The



MODIS data are accessible at http://giovanni.gsfc.nasa.gov/aerostat/. The Terra visible images
were obtained at https://worldview.earthdata.nasa.gov/. The hourly PBL height used was based
on the Modern-Era Retrospective analysis for Research and Applications, Version 2 (MERRA2)
reanalysis data. The severe haze days were selected with daily $PM_{2.5}$ (particular matter with
aerodynamic diameter less than 2.5 micron) concentration greater than 200 μg m$^{-3}$, and the
typical clean days were limited to the days with daily $PM_{2.5}$ concentration smaller than 30 μg
m$^{-3}$. The PBL height and the $PM_{2.5}$ surface concentration at 14:00 Beijing time (BJT) each day
in 2013 were used for the correlation analysis. All raining days were filtered out when
analyzing the correlation between the PBL height and the $PM_{2.5}$ concentration. The surface
solar radiation (SSR) data were based on the satellite retrievals (Tang et al., 2016b), which are
accessible at http://www.tpedatabase.cn.

Ground-based measurements of fine particulate matter or $PM_{2.5}$ employed in our

analysis covered the period from 25 September to 14 November 2013. The hourly $PM_{2.5}$
concentrations in Beijing (BJ) were obtained from the Embassy of United States in Beijing
(http://www.stateair.net/web/historical/1/1.html). The $PM_{2.5}$ mass concentrations in Baoding
(BD) and Shijiazhuang (SJZ) were obtained from https://air.cnemc.cn:18007/. Measurements
of PM properties in Beijing were taken from that previously reported by Guo et al. (2014),
which provided $PM_{2.5}$ concentration, aerosol chemical composition, and gaseous data for
correlation analysis and constrains for modeling studies. The observation-based analysis and
the modeling study focused on two severe haze episodes, i.e., 25 September – 30 September
(episode 1 or EP1) and 2 October – 6 October (episode 2 or EP2), 2013 in NCP.
**2.2. Model experiments**
**2.2.1. Simulations on the haze-PBL interactions**

The aerosol-PBL interactions during the severe haze events and the associated regional

climate effects were examined by conducting WRF modeling sensitivity studies. An aerosol

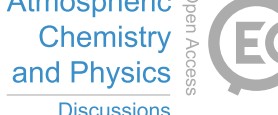

radiative module was implemented by Fan et al. (2008) to the Goddard Shortwave Radiation
Scheme to online compute the wavelength-dependent aerosol optical properties, including the
AOD, the asymmetry factor and the single scattering albedo (SSAs). Aerosol particles with the
core-shell configuration in the aerosol radiative module were assumed to consist of BC (core)
and ammonia sulfate (shell). The hygroscopic growth of aerosol particles was taken into
account, following Mallet et al. (2004). A two-moment bulk microphysical scheme developed
by Li et al. (2008) was employed, which has been widely used to investigate the aerosol-cloud
interactions under various cloud systems (Wang et al., 2014b; Wang et al., 2014a; Lin et al.,
2016). A 100×100 grids domain with a horizontal grid spacing of 2 km and 50 vertical levels
with stretched grid spacings was set up to cover the entire urban region of Beijing. The initial
and boundary meteorological conditions were generated from six-hourly NCEP FNL (Final)
Operational Global Analysis (1°×1°). No convective parametrization was applied for the
simulations.

We performed simulations on the two haze episodes (EP1 and EP2). The two days prior

to the two haze episodes (25 September and 2 October, 2013) are denoted as the clean periods,
while the most polluted days during the two episodes, i.e., 28 September and 5 October are
denoted as the polluted periods. The aerosol number size distributions for initial and boundary
conditions of all simulation were based on the measurements during the 2013 field campaign
at Beijing (Fig. S1). The aerosol measurements on 25 September and 2 October 2013 were
taken as the input for the clean cases and 28 September and 5 October 2013 for the polluted
cases. The aerosol surface number and mass concentration for modeling initialization were set
as $3.5×10^4$ cm$^{-3}$ ($3.6×10^4$ cm$^{-3}$) and 10 μg m$^{-3}$ (11 μg m$^{-3}$) for the clean case of EP1 (EP2) and
$1.7×10^4$ cm$^{-3}$ ($1.8×10^4$ cm$^{-3}$) and 280 μg m$^{-3}$ (310 μg m$^{-3}$) for the polluted case of EP1 (EP2),
respectively, consistent with the field measurements. Also, based on the measurements, the BC
percentage in total aerosol mass was set as 10.0% and 6.0% for the clean and polluted cases,





respectively. The two polluted days for simulations were cloud-free days, therefore the aerosol
indirect effects were ruled out.

To assess the role of BC in the aerosol suppression effect on the PBL development and

the aerosol radiative forcing during haze evolution, we performed a set of sensitivity
simulations under the polluted condition by excluding the BC effects (referred as non-BC case),
in which the BC radiative effect was turned off by assigning a zero value to the real and
imaginary parts of BC refractive index, i.e., the SSA in non-BC case was equal to unity. To
quantify the BC aging effects, additional simulations were carried out for fresh BC (denoted
by fresh-BC), in which the BC core was not imbedded in the non-BC shell and the optical
parameters for the BC and non-BC components were calculated separately by the Mie theory.
In the fresh-BC case, the lensing effect due to the coating during the aging process was
excluded, but the restructuring effect induced by aging was considered partially since the BC
core was assumed to be spherical and in the compact shape. Alternatively, a case for aged BC
(denoted by aged-BC) was treated by considering the full aerosol components (with both BC
and non-BC components) and the core-shell configuration. A summary of the simulation cases
is listed in Table 1.

One deficiency to predict the absorbed AOD and the directive radiation forcing of BC

in atmospheric models is relevant to the underestimation in coating-enhancement of BC
absorption (Bond et al., 2013). To assess the potential bias on the radiative effects of aged BC,
additional simulations on the polluted conditions were conducted by constraining the
enhancement of mass absorption cross section of BC ($E_{MAC-BC}$) according to the experimental
value, i.e., 2.4, derived from a chamber study in Beijing (Peng et al., 2016). Though the $E_{MAC-BC}$
of 2.3 derived from the aged-BC case is slightly lower than that reported by Peng et al.
(2016), comparison between the two simulations indicates little difference in the
thermodynamic/dynamic conditions and the radiative budget.





**2.2.2. Empirical estimation of the moisture effect on haze-PBL interactions**
In addition to the numerical model simulations, we employed an empirical equation
derived by Nozaki (1973) and modified by Tie et al. (2017) to examine the RH sensitivity in
the boundary layer to the PBL height based on observed meteorological conditions:
$H = \frac{121}{6}(6 - P)(T - T_d) + \frac{0.169P(U_Z + 0.257)}{12f \ln Z/z_0}$     (1)
where $H$, $T$, $T_d$, and $U_z$ represent the PBL height (m), surface air temperature (K), surface
point (K), and mean wind speed (m s$^{-1}$) at height of $Z$ ($Z$=10 m), respectively. $f$ and $z_0$ are the
Coriolis parameter (s$^{-1}$) and surface roughness length (0.5 m in this study), respectively. $P$ is
the Pasquill stability level, classified as six categories from very unstable (A), moderately
unstable (B), slightly unstable (C), neutral (D), slightly stable (E) to moderately stable (F)
(Pasquill, 1961). To relate RH with the PBL height, we adopted a modified Nozaki's equation
using (100- RH)/5 to replace ($T$-$T_d$) according to Wallace and Hobbs (2005) and Tie et al.
(2017). The measured wind speeds were used in the calculations. For the severe haze events,
the atmosphere was stable, and the Pasquill stability levels were set as 4~5. The input of the
PBL height for the aged-BC cases were based on ceilometer measurements, and we increased
the PBL height by about 1000 m as the input for the clean cases, which was estimated based
on MERRA2 reanalysis data due to the lack of relevant measurements.
**3. Results and Discussion**
**3.1. Regional characteristics of severe haze episodes**
Measurements of the PM$_{2.5}$ mass concentrations from 25 September to 14 November
2013 reveal that severe haze occurs frequently over the NCP, reflected by a periodic cycle of
4-7 days with highly elevated PM pollution (Fig. 1a-c). Each severe haze episode consists of a
clean period, a transition period from clean to hazy conditions, and a polluted period with very
high PM levels. For the three megacities across the NCP, i.e., Beijing, Baoding, and



Shijiazhuang, the maximal mass concentration of $PM_{2.5}$ consistently exceed several hundred
$\mu g\ m^{-3}$ during the polluted period. The $PM_{2.5}$ concentrations at the three megacities exhibit a
remarkable similarity in the timing and magnitude for the peak $PM_{2.5}$ concentrations. The
nearly synchronized temporal variations in the PM levels among the three megacities indicate
a prominent characteristic of severe haze formation, indicating the importance of *in-situ* PM
production over the entire region. During the evolution from clean, transition, to polluted
periods, the RH and wind speed is consistently increased and decreased, respectively (Fig. 1d).

The two polluted events on 28 September (EP1) and 5 October (EP2) are captured from

both *in-situ* measurements (Fig. 1) and satellite observations (Fig. 2). The satellite MODIS data
illustrate that the maximal AOD area occurs in the three megacities (i.e., BJ, BD, and SJZ). For
example, the AOD value in Beijing exceeds 4.0 and 2.0, during EP1 and EP2, respectively.
The spatial distribution of severe regional haze events is also depicted from the satellite visible
images, showing that a grey haze plume covers a substantial portion of the NCP region (Figs.
2c and d). The coincidence of the highest AOD areas with the locations of the megacities is
also discernable from the mean AOD values averaged over all the hazy days (e.g., daily $PM_{2.5} >$
$200\ \mu g\ m^{-3}$) in 2013 (Fig. S3), showing a large zone of elevated AOD values over the three
megacities. In contrast, the fall seasonal and annual AOD means averaged over all days in 2013
show that the maximal AOD values are located to the south of Beijing (Figs. 2e and f),
reflecting the typical regional transport pattens over this region (Guo et al., 2014; An et al.,2019;
Peng et al., 2021). In addition, the occurrence of severe haze events is consistently
accompanied by stagnant weather, characterized by weak southerly winds in Beijing and its
surrounding areas (Figs. 2a and b). For example, the wind speed is typically less than $1\ m\ s^{-1}$
in the highest AOD area (Figs. 2a and b), compared to that of a few to ten $m\ s^{-1}$ during the clean
period (Fig. 1d). Air stagnation retards PM dispersion, resulting in minimal regional transport
during the polluted period. On the other hand, the gaseous aerosol precursors (e.g., $SO_2$, $NO_x$,


VOCs, and NH$_3$ with the chemical lifetimes from hours to days) are sufficiently transported
and dispersed prior to haze development over this region, as evident from much higher wind
speeds during the clean period (Fig. 1d). Efficient regional transport of the gaseous aerosol
precursors explains the similarity in the spatial/temporal PM variations, since well-mixed
gaseous aerosol precursors result in similar *in-situ* PM production under stagnant conditions
(Figs. 1a-c). Moreover, the coincidence of the AOD hotspots with the three megacities (Figs.
2a,b and S3) indicates more efficient *in-situ* PM production over the megacities, suggesting a
key role of traffic emissions (i.e., anthropogenic VOCs and NO$_x$) in facilitating regional severe
haze formation. While wind fluctuation likely results in PM variation in an isolated location,
especially for Beijing, which is situated at the northern edge of the NCP (Li et al. (2015), our
analysis of temporal/spatial PM distributions indicates that the dominant regional features
during the polluted period are reflected by rapid *in-situ* PM production and inefficient transport,
both of which are amplified by air stabilization.
**3.2. Photochemical PM formation**

To further elucidate the role of *in-situ* photochemical production in haze development,

we analyzed the temporally resolved PM properties in Beijing. Evidently, the PM$_{2.5}$ mass
concentration increases by more than 200 μg m$^{-3}$ in less than 8 hrs. during the transition period
for EP1 and EP2 (Figs. 3a and b), which is dominated by the increase in the SOA mass
concentration linked to photochemical oxidation of VOCs (Guo et al., 2014; Liu et al., 2021).
The mass concentration of oxygenated organic aerosol (OOA) is typically considered as a
surrogate for SOA (Wood et al., 2010). Since OOA and the level of oxidants, O$_x$ ([$O_x$] $\equiv$
[$O_3$] + [$NO_2$]), are both produced from oxidation of VOCs (Suh et al., 2001; Fan and Zhang,
2004; Zhao et al., 2005) and have a lifetime of longer than 12 hours, it is anticipated that both
quantities are correlated, when their formations occur on a similar timescale and at the same
location (Atkinson, 2000). Figs. 3a-d show that the increase in OOA is well correlated with the



$O_x$ level during the transition period. The $R^2$ from linear regression between OOA and $O_x$
during the transition period (i.e., from 7:00 am to 2:30 pm) is 0.96 for EP1 and 0.95 for EP2
(Fig. 3e). The high correlation between OOA and $O_x$ implies important *in-situ* production of
PM via photochemical reactions, consistent with the ground-based measurements (Fig. 1) and
satellite observations for PM (Fig. 2 and Fig. S3). The mean ratio of [OOA] to [$O_x$] for the two
episodes in Beijing is 0.34 ($\mu g\ m^{-3}\ ppb^{-1}$), suggesting highly efficient photochemistry. For
comparison, the mean ratio of [OOA] to [$O_x$] during the two episodes in Beijing is about 2.4
and 5.1 times of those in Mexico City and Houston (Wood et al., 2010), respectively, indicating
that the photochemical PM formation in Beijing is much more efficient than those in Mexico
City and Houston (Fig. 3f). The more efficient photochemical formation of PM in Beijing is
attributable to the presence of higher levels of anthropogenic aerosol precursors, such as
anthropogenic VOCs and $NO_x$, than those in the other two cities (Guo et al., 2014; Zhang et
al., 2015a). On the other hand, the correlation between [OOA] and [$O_x$] exists only during the
transition stage but vanishes during the polluted period. The latter is evident from the
continuing increase in [OOA] but decreasing [$O_x$]. In particular, $O_3$ production is significantly
suppressed during the polluted periods because of reduced solar ultraviolet radiation, leading
to inefficient photooxidation (Wu et al., 2020; Peng et al., 2021). Several previous studies have
attributed highly elevated levels of $PM_{2.5}$ during the polluted period to the importance of
multiphase chemistry to contribute to SIA and SOA formation (Wang et al., 2016; An et al.,
2019; Peng et al., 2021). For example, sulfate formation is effectively catalyzed by BC (Zhang
et al., 2020) and considerably enhanced via aqueous oxidation of $SO_2$ by $NO_2$ in the presence
of $NH_3$ during the transition/polluted periods (Wang et al., 2016), both increasing with
increasing RH. Also, oligomerization from dicarbonyls increases at high RH (Li et al., 2021a,
b), contributing to significantly enhanced SOA formation during the polluted periods (Zhang
et al., 2021).





### 3.3. Impacts of the haze-PBL interaction

### 3.3.1. A positive feedback of PM accumulation

To assess the impacts of haze-PBL interactions on PM pollution, we evaluated the correlation between the PM level and PBL height. Fig. 4 shows an analysis of daily mean PBL height versus $PM_{2.5}$ concentration between clean and hazy days from the ground-based measurements and the MERRA2 reanalysis data in 2013. The daily maximal PBL height is negatively correlated with surface $PM_{2.5}$ concentration (Fig. 4a). The diurnal cycle of the PBL height shows that the PBL height on severe haze days (daily $PM_{2.5}$ concentration > 200 μg m$^{-3}$) is significantly lower than that on clean days (daily $PM_{2.5}$ concentration < 30 μg m$^{-3}$), with a maximum difference of 800 m (Fig. 4b). Furthermore, the dimming area over NCP, which is reflected by the lower mean of the satellite-retrieved surface solar radiation (SSR) averaged over all the severe haze days in 2013, coincides with the region with the highest AOD (Fig. S3), implying a strong spatial association between the solar radiation intensity and PM pollution at the surface. The co-locations in the areas between the lowest SSR and highest AOD also reflects the occurrence of the highest PM levels at the megacities during the regional severe haze episodes.

We further elucidated the response of PBL development to the PM pollution, and the linkage between the aerosol-PBL interactions and aerosol radiative effects are further elucidated by performing sensitivity modeling studies on the two hazy days (Figs. 5-6). The performance of the model simulations was validated by comparison with field observations. The simulated temperature and RH are consistent with the sounding data in light of the vertical variations (Fig. S2). The simulated AOD at 550 nm is 0.05 and 3.6 on 25 and 28 September 2013 for EP1, respectively, and 0.04 and 2.0 on 2 and 5 October 2013 for EP2, respectively, in qualitative agreement with the Aerosol Robotic Network (AERONET) measurements in Beijing (Table 2). The simulated one-day accumulated surface solar radiation and the peak



solar radiation flux in the aged-BC case for EP1 (EP2) are 9.2 MJ m$^{-2}$ (11.3 MJ m$^{-2}$) and 326
W m$^{-2}$ (402 W m$^{-2}$), respectively, comparable to the ground-based measurement of 10.6 MJ m$^{-}$
$^2$ (9.8 MJ m$^{-2}$) and 408 W m$^{-2}$ (452 W m$^{-2}$) in Table 2. The temporal evolutions of PBL and its
peak heights derived from the aged-BC cases are also consistent with the available
measurements (Figs. 5a and g).

The simulated maximal height of PBL under the polluted condition is reduced by more

than 300 m relative to the clean condition (Figs. 5a and g). The reduction in PBL height is
explained by the aerosol radiative effects. Under the polluted condition, a warmer temperature
is located at the altitude of around 1.2 km, and less SSR reaches the ground level (Figs. 5e and
k, f and l). Also, the surface temperature is reduced by several degrees (Figs. 5e and k, and
Figs. 6a and c). Consequently, the turbulent kinetic energy (TKE) is reduced, and the updraft
is weakened in the aged-BC cases relative to the clean cases (Figs. 5b, c, h and i), leading to an
enhanced atmospheric stratification and hindered development of PBL. The largely reduced
TKE during the polluted periods from the model simulations is consistent with field
measurements, showing that the turbulent fluxes are greatly reduced in the mixed surface layer
under polluted conditions (Wilcox et al., 2016). In addition, surface winds are reduced by 0.7
m s$^{-1}$ from clean to aged-BC cases (Fig. 6b and d), leading to suppressed entrainment aloft and
restricted development of the PBL.

The interaction between aerosols and PBL induces further feedbacks at the surface by

altering atmospheric dynamic/thermodynamic conditions and stability. For example, the PM
concentration at the ground level accumulates when the PBL is compressed, resulting in a
smaller extent for vertical dilution. Also, the diurnal feature of PM pollution diminishes
because of collapsed PBL, allowing PM to continuously accumulate at the surface. In addition,
horizontal advection is also suppressed under polluted conditions, as reflected by weak wind
speeds. Consequently, the heavy haze period persists over an extensive period (about 4-7 days)





over this region and is only dissipated by strongly northly winds associated with frontal passage
(Guo et al., 2014; An et al., 2019). The continuous PM accumulation for multiple days over
the NCP is distinct from other megacities across the world, such as Houston, Los Angeles, and
Mexico City, which always exhibit a clear diurnal feature of the PM levels (Zhang et al., 2015a),
implying a key role of the haze-PBL interaction in deteriorating air quality and worsening the
hazy condition in this region.

The suppression in PBL height results in significant enhancement of atmospheric

moisture, another crucial factor affects the haze evolution, which promotes the occurrence of
multiphase reactions (Li et al., 2021a, b). The measured RH increases greatly during the two
episodes (Fig. 1d), i.e., from about 18%-19% on the clean days (25 September and 2 October)
to 53%-55% on the polluted days (28 September and 5 October). To evaluate the sensitivity of
the atmospheric moisture to the PBL height, we employed a modified Nozaki's equation
(Nozaki, 1973; Tie et al., 2017) to calculate the RH under different PBL height scenarios using
the observed meteorological conditions as inputs (Table 3). The calculated RH increases from
19% to 68% for EP1 and from 21% to 73% for EP2, when the PBL height decreased by about
1000 m during the polluted days, indicating that the humidity is highly sensitive to the PBL
height.

The elevated RH during the polluted period is explained from collapsed PBL to inhibit

vertical moisture transport, reduced surface temperature leading to lower saturation vapor
pressure, and inefficient entrainment of dry air aloft (Fan et al., 2008; Liu et al., 2013). In
addition, enhanced moisture leads to hygroscopic growth of aerosol particles (Liu et al., 2013;
Tie et al., 2017). For example, the growth hygroscopic factor relevant to the RH enhancement
during EP1 and EP2 increases from 1.3 on the clean days to 1.5 on the hazy days, using an
empirical equation derived according to Meier et al. (2009). The additional aerosol growth
causes additional attenuation of incoming solar radiation by scattering and absorption to


416 amplify PBL suppression. Moreover, an enlarged aerosol surface area (due to hygroscopic

417 growth) and elevated RH during the polluted periods favor aqueous-phase reactions to produce

418 sulfate, nitrate, and SOA (Wang et al., 2016a). For example, a recent experimental/field study

419 has shown enhanced sulfate formation, which is catalyzed by BC and increases monotonically

420 from 10% to 70% RH (Zhang et al., 2020). Also, the aqueous reaction of dicarbonyls, which

421 are produced with high yields from oxidation of aromatic VOCs, is significantly enhanced at

422 high RH to yield oligemic products and enhance SOA formation (Li et al., 2021a; b). Hence,

423 enhanced PM production near the ground level strengthens the suppressing effect for the PBL

424 development and results in stabilization and moisture enhancement, constituting positive

425 feedback to amplify the haze development.

426 **3.3.3. The BC effects**

427 We performed model sensitivity simulations to elucidate the role of BC in PBL

428 suppression by considering the non-BC, fresh-BC, and aged-BC scenarios during the polluted

429 periods. Comparison shows a negligible effect on the haze-PBL interaction between the non-

430 BC and fresh-BC cases (Figs. 5, 6 and S4) but large changes in solar radiation and

431 thermodynamic/dynamic conditions within the PBL between the non-BC/ fresh-BC and aged-

432 BC cases, which are attributed to the radiative effects of aged BC. For example, the shortwave

433 heating rate per unit mass is much larger for aged-BC than non-BC and is two times higher for

434 aged-BC than fresh-BC (Figs. 5d and j), suggesting that the BC aging process greatly attenuate

435 incoming solar radiation. Although BC accounts for only 6% of the total aerosol mass under

436 the polluted conditions, about one third of the total reduction in SSR for full-component

437 aerosols is attributed to absorption enhancement after BC aging (Figs. 5f and l). The reduced

438 SSR by the BC aging leads to a cooling of 0.5-0.8 K at the surface. As a result, BC aging

439 contributes significantly to atmospheric stabilization, as evident from weaker updrafts, smaller

440 TKE, and shallower PBL for the aged-BC case (Fig. 5).



The BC aging causes a decrease in the maximum PBL height (at noontime) by about
150 m for the aged-BC case compared to the non-BC and fresh-BC cases. Overall, the BC
aging contributes more than 30% of the total reduction in the PBL height by all aerosol
components. The restricted PBL development by BC absorption in our work is consistent with
that identified previously (Ding et al., 2016; Petäjä et al., 2016). Using a radiative transfer
model, Zhang et al. (2020) shows large strongly positive radiative forcing in the atmosphere
and strongly negative radiative forcing at the surface by BC aging, consistent with those of the
maximal estimates at about noontime from our calculations (Fig. 5i,f).
**3.5. Aerosol direct radiative forcing**
The aerosol direct radiative forcing during regional haze also exhibits a profound
climatic effect (Ramanathan et al., 2007). Fig. 7 shows that the total aerosol radiative forcing
at the surface (SFC) and in the atmosphere (ATM) during the haze episodes EP1 (EP2) are -
87.8 (-62.8) W m$^{-2}$ and 82.2 (56.9) W m$^{-2}$, respectively. The positive radiative forcing by all
aerosols in the atmosphere is dominated by that of aged BC, which accounts for 80% of the
total radiative forcing for both episodes. The net radiative forcing at the top of the atmosphere
(TOA) by all aerosols for EP1 (EP2) is around -5.6 (-5.9) W m$^{-2}$, much smaller than the non-
BC case with a large negative value of -36.8 (-26.0) W m$^{-2}$. The strong cooling at the surface
is largely canceled out by the strong warming in the atmosphere under the polluted condition,
leading to a small net TOA forcing.  Clearly, BC aging contributes significantly to cooling at
the surface and warming aloft and, hence, the overall radiative budget during the polluted
periods. Climatologically, the aerosol TOA forcing on the regional/national level has been
shown to be nearly zero or slightly positive in China (Li et al., 2007; Ramanathan et al., 2007;
Ding et al., 2016), also demonstrating that the large positive forcing by absorbing aerosols
greatly compensates the negative forcing by the non-absorbing aerosols (Table S1). Therefore,
regional global warming is likely mitigated by reducing BC emissions (Wang et al., 2015b).





### 4. Conclusions


In this work, we analyzed the temporal and spatial characteristics of PM pollution
during severe haze events over NCP, by examining ground-based measurements and satellite
observations. Severe haze occurs frequently over this region, evident from a periodic (4-7 days)
cycle of highly elevated PM pollution. The PM evolutions among the three megacities (Beijing,
Baoding and Shijiazhuang) exhibit a remarkable similarity during the haze events, showing
nearly synchronized temporal variations in the PM levels. The similar timing and magnitude
in the peak $PM_{2.5}$ concentrations among the three megacities indicate significant *in-situ* PM
production. Satellite measurements show that the AOD hotspots during the polluted period are
co-located with the three megacities, but are distinct from seasonal and annual AOD means,
indicating the importance of urban emissions (mainly traffic emissions consisting of
anthropogenic VOCs and $NO_x$). *In-situ* PM production occurs most efficiently over the
megacities, and urban sources relevant to traffic emissions play a critical role in regional severe
haze formation.
Our result reveals that the rapid photochemistry drives the PM production during the
transition period. There exist concurrent increases in OOA and $PM_{2.5}$ concentrations and a
strong correlation between OOA and $O_x$ concentrations during this period. The [OOA]/[Ox]
ratio in Beijing is much higher than that in Mexico City and Houston, attributable to much
higher level of gaseous precursors (i.e., anthropogenic VOCs and $NO_x$) in Beijing than the
other two cities. The correlation between [OOA] and [$O_x$], however, vanishes during the
polluted period, when $O_3$ production is significantly suppressed because of reduced solar
ultraviolet radiation and inefficient photooxidation (Wu et al., 2020; Peng et al., 2021). The
continuing increases in $PM_{2.5}$ and OOA with decreasing $O_x$ during the polluted period implies
a key role of multiphase chemistry in driving the haze severity, when the RH level is
significantly elevated. The continuous growth in $PM_{2.5}$ and OOA during the polluted period





has been explained by an increasing importance of heterogeneous chemistry to contribute to
sulfate, nitrate, and SOA formation (Wang et al., 2016a; An et al., 2019; Peng et al., 2021;
Zhang et al., 2021).

Using the WRF model coupled with an explicit aerosol radiative module, we elucidated

the underlying mechanism relevant to the haze-PBL interactions, showing a positive feedback
to haze formation at the ground level. The PBL height is largely reduced under the polluted
condition, since the PBL is markedly suppressed (as indicated by the reduced TKE and
weakened updraft), because of strong aerosol heating in the atmosphere and strong cooling at
the surface. The PM concentration near the surface accumulates significantly in a compressed
PBL, since PM dispersion is unfavorable in the stratified and collapsed PBL, leading to
continuous growth and accumulation of PM over multiple days. Calculations using the
modified Nozaki's equation shows that the suppressed PBL results in a great enhancement of
atmospheric moisture near the surface. A more humid condition leads to hygroscopic growth
of aerosol particles and more efficient multiphase PM production. Therefore, haze development
near the surface is considerably exacerbated because of the positive feedback in responding to
the atmospheric moisture and thermodynamic/dynamic conditions to amplify the haze severity.

Our combined observational analysis of the temporal/spatial PM distributions and

modeling unravel a dominant regional characteristic for severe haze evolution in the NCP
region, showing rapid *in-situ* PM production and inefficient transport, both of which are
amplified by air stabilization. On the other hand, regional transport sufficiently disperses the
gaseous aerosol precursors ($SO_2$, $NO_x$, VOCs, and $NH_3$) during the clean period, which
subsequently result in rapid *in-situ* PM production via photochemistry during the transition
period and via multiphase chemistry during the polluted period.

The modeling simulations on two haze episodes indicate important regional climatic

effects. The net TOA forcing for the two hazy days is about of -5.6 ~ -5.9 W m$^{-2}$, showing



strong negative radiative forcing (cooling) of -63 to -88 W m$^{-2}$ at the surface and strong positive
radiative forcing (warming) of 57 to 82 W m$^{-2}$ in the atmosphere. BC represents the dominant
contributor to the positive aerosol radiative forcing in the atmosphere, thus playing a significant
role in the haze-PBL interaction. Specifically, BC aging contributes to more than 30% of the
PBL collapse induced by total aerosols and about 50% of the TOA positive radiative forcing.
Our work highlights the necessity to better understand the BC aging process and improve
representation in atmospheric models for accurate assessment of the aerosol climatic effects.
We conclude that reduction in BC emissions achieves co-benefits, which improve local and
regional air quality by minimizing air stagnation and mitigate the global warming by alleviating
the positive direct radiative forcing.
**Acknowledgement**
This work was partially supported by a collaborative Program between the Texas A&M
University (TAMU) and the Natural Science Foundation of China (NSFC). R.Z. acknowledged
additional support by the Robert A. Welch Foundation (Grant A-1417). The modeling portion
of this research was conducted at the TAMU High Performance Research Computing. We
thanked Hong-Bin Chen and Philippe Goloub for the data at the Beijing AERONET site.



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

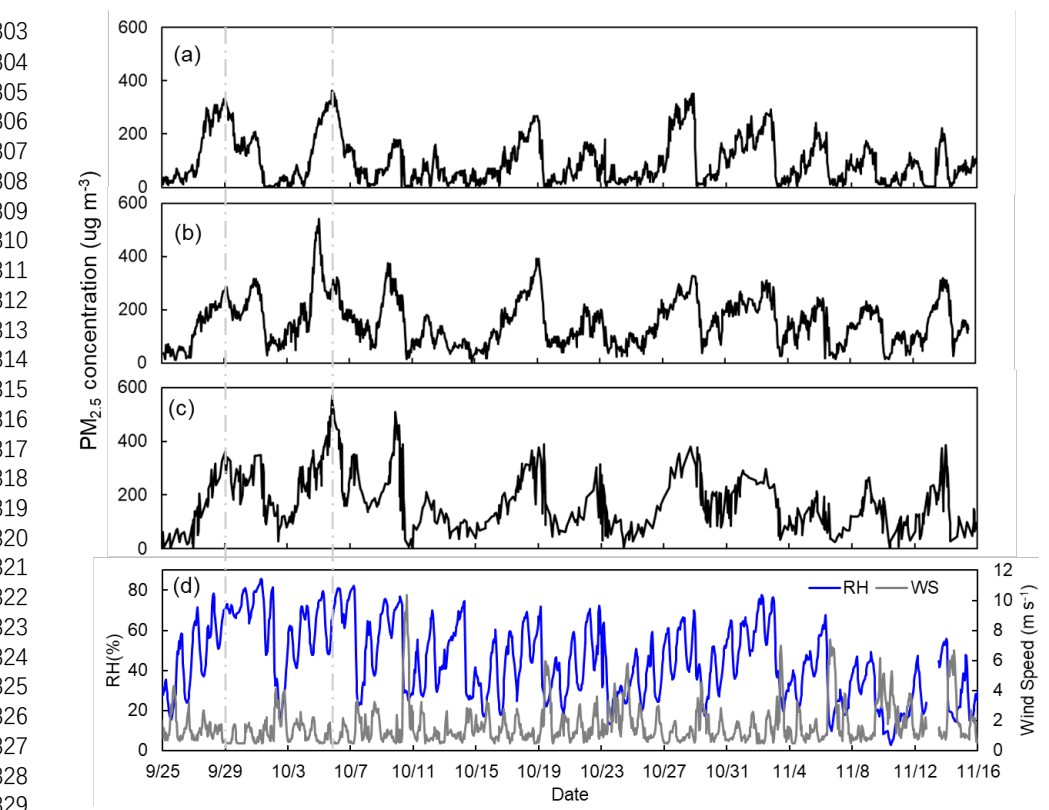

**Figure 1.** Time series of PM$_{2.5}$ mass concentration measured at three megacities over North China Plain (NCP), including (a) Beijing, (b) Baoding, and (c) Shijiazhuang from 25 September to 16 November, 2013, and (d) the associated relative humidity (RH, blue line) and 10-m wind speed (grey line) in Beijing. The PM$_{2.5}$ mass concentration and meteorological fields in Beijing are taken from Guo et al. (2014), and the PM$_{2.5}$ data for Baoding and Shijiazhuang are taken from https://air.cnemc.cn:18007/. Two severe haze episodes from 25-29 September and from 2-7 October are selected as the case studies in this work, and the two vertical dash lines label the time for the peak PM$_{2.5}$ concentration in Beijing.

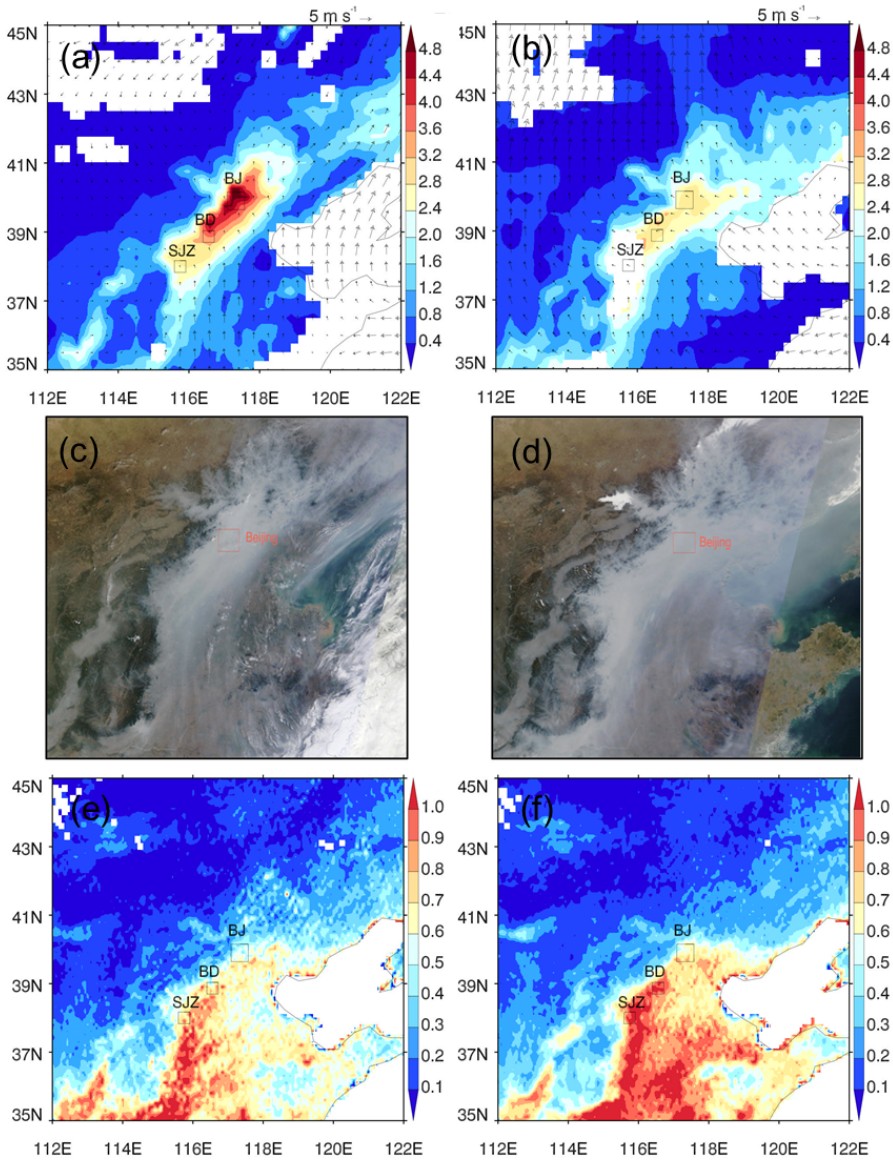

839
840
**Figure 2.** MODIS AOD (a-b) and visible images (c-d) illustrating the two severe haze episodes
in Fig. 1. (a) and (c) correspond to 28 September, 2013, and (b) and (d) correspond to 5 October,
2013. (e) and (f) represent MODIS AOD of fall seasonal and annual mean in 2013. The
megacities of Beijing (BJ), Baoding (BD) and Shijiazhuang (SJZ) are marked as squares. Wind
field imposed on (a) and (b) is based on ECMWF reanalysis data.



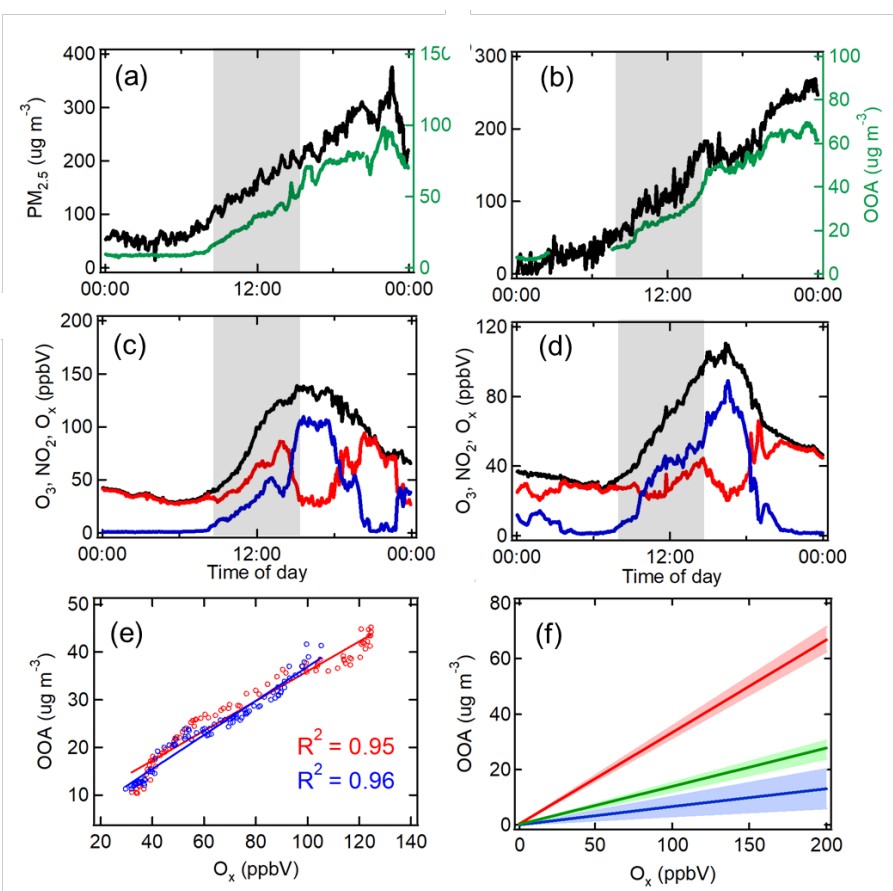

**Figure 3.** The temporal evolutions of measured PM$_{2.5}$ (black) and OOA (green) mass concentrations (a-b) and O$_3$ (blue), NO$_2$ (red), and O$_x$ (black) mixing ratios (c-d) during the early stages of the two haze episodes. (a) and (c) are for the episode starting on 27 September, 2013, and (b) and (d) are for the episode starting on 4 October, 2013. (e) represents linear regression between O$_x$ and OOA on 27 September (red circles) and 4 October (blue circles), 2013, using measurements at the transition periods (grey shadings (a-d)). (f) corresponds to the ratios of [OOA] changes to [O$_x$] changes (Δ[OOA]/Δ[O$_x$]) for Beijing (red), Mexico City (green) and Houston (blue). The ratios for Beijing are derived from this study, and the ratios for Mexico City and Houston taken from Wood et al. (2010). Color shadings in (f) represent the range between the minimum and maximum ratios.


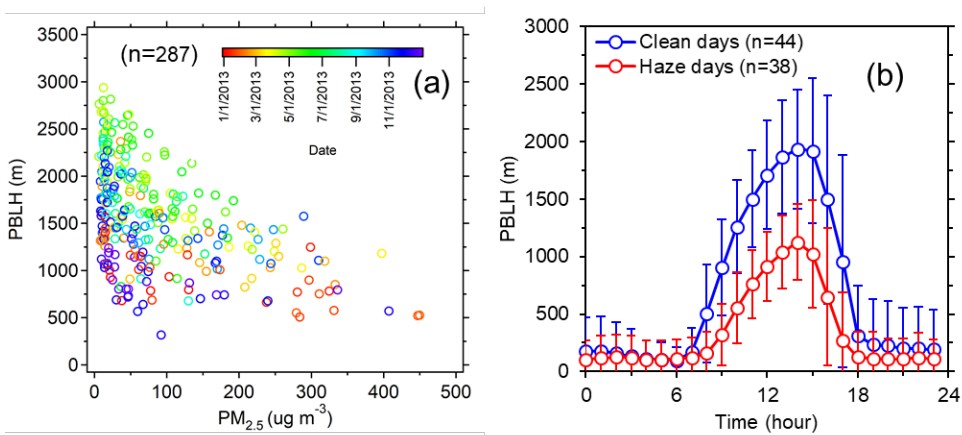

857

**Figure 4.** (a) Scattering plot for daily mean PBL height versus $PM_{2.5}$ concentration and (b) mean diurnal variations of PBL height averaged over clean days (daily mean $PM_{2.5} < 30$ μg m$^{-3}$) and extremely hazy days (daily mean $PM_{2.5} > 200$ μg m$^{-3}$) in 2013 at Beijing, China. *n* denotes the number of days used for plotting. The vertical lines in (b) denote $\pm 1$ standard deviation. All the precipitation days were filtered out.

863

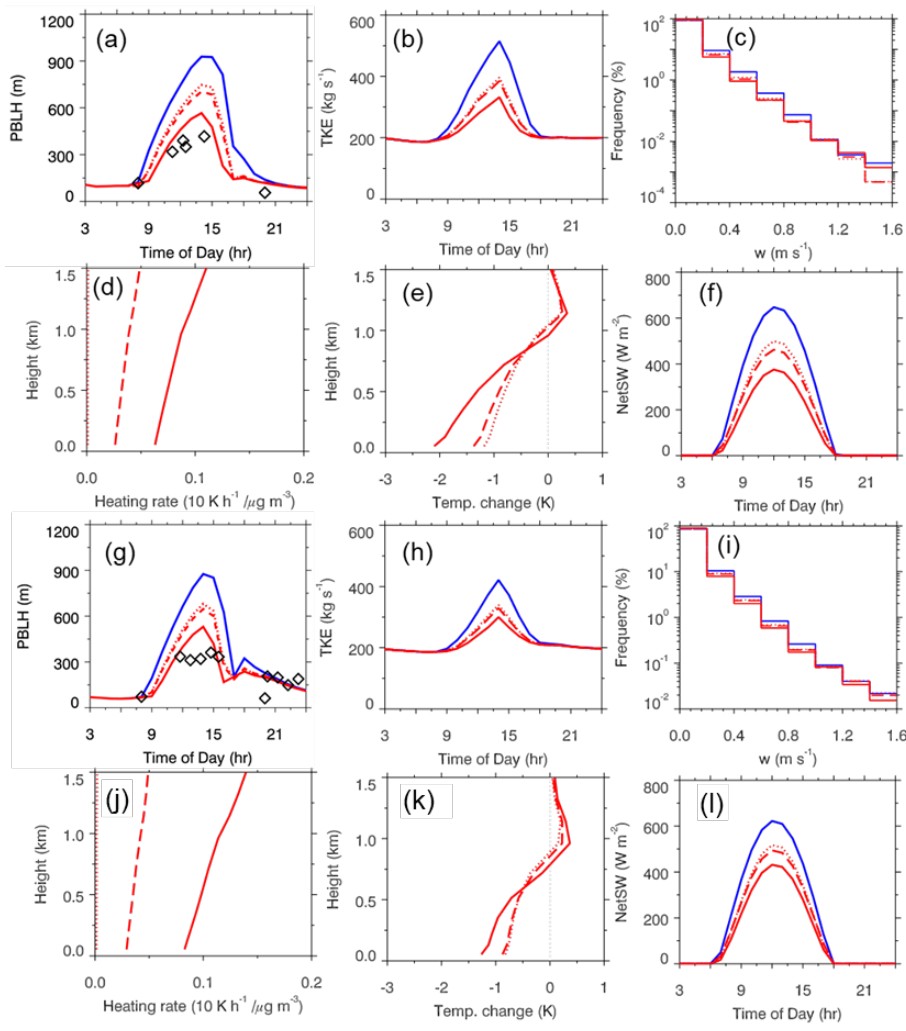

864

**Figure 5.** Simulated meteorological conditions and thermodynamic and dynamic feedbacks
under the clean conditions (blue solid) and the polluted conditions for the non-BC (red dot),
fresh-BC (red dashed), and aged-BC (red solid) cases. (a) and (g) correspond to simulated
diurnal variations of PBL height, (b) and (h) correspond to the diurnal variations of vertically
integrated TKE, (c) and (i) represent the frequency distribution of updraft. (d) and (j) are the
vertical profile of the shortwave heating rate per unit aerosol mass for the non-BC (red dot
line), fresh-BC (red dash line), and aged-BC (red solid line) cases. (e) and (k) are similar as
(d) and (j) but for the temperature changes. (f) and (l) are diurnal evolutions of net surface
shortwave radiation (NetSW). (a-f) are for EP1 and (g-n) are for EP2. The black hollow
squares in (a) and (g) denote measurements of PBL height from ceilometer.

875





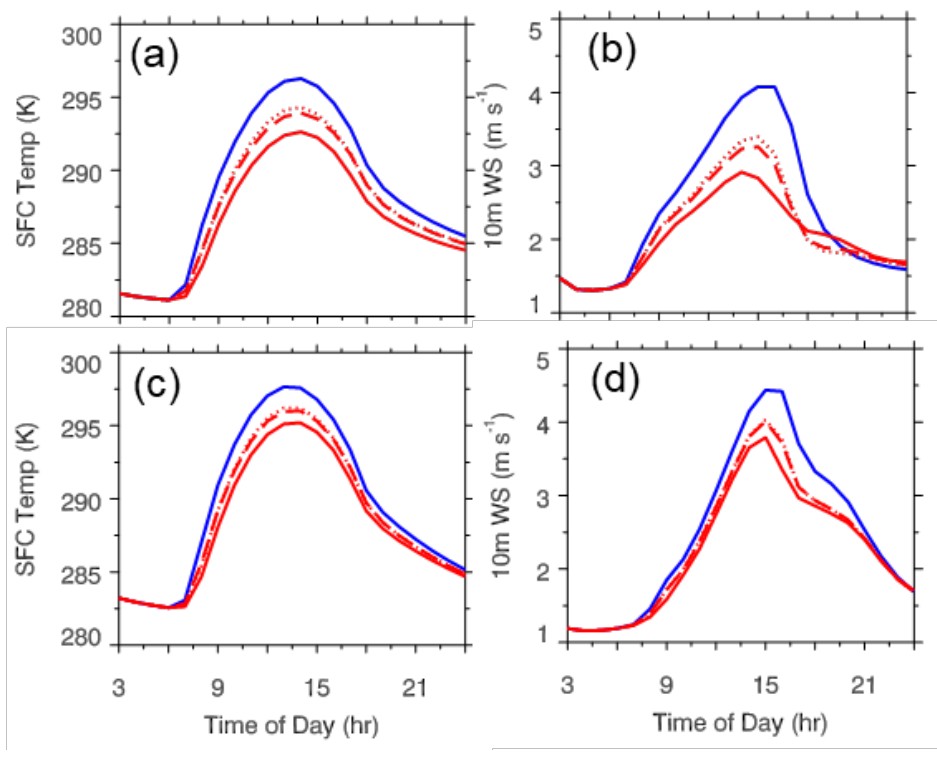

876

**Figure 6.** Temporal evolutions of surface temperatures (a and c) and 10-meter wind speeds (b and d) under the clean conditions (blue solid) and the polluted conditions for the non-BC (red dot), fresh-BC (red dashed), and aged-BC (red solid) cases. (a) and (b) correspond to EP1, and (c) and (d) correspond to EP2.

881


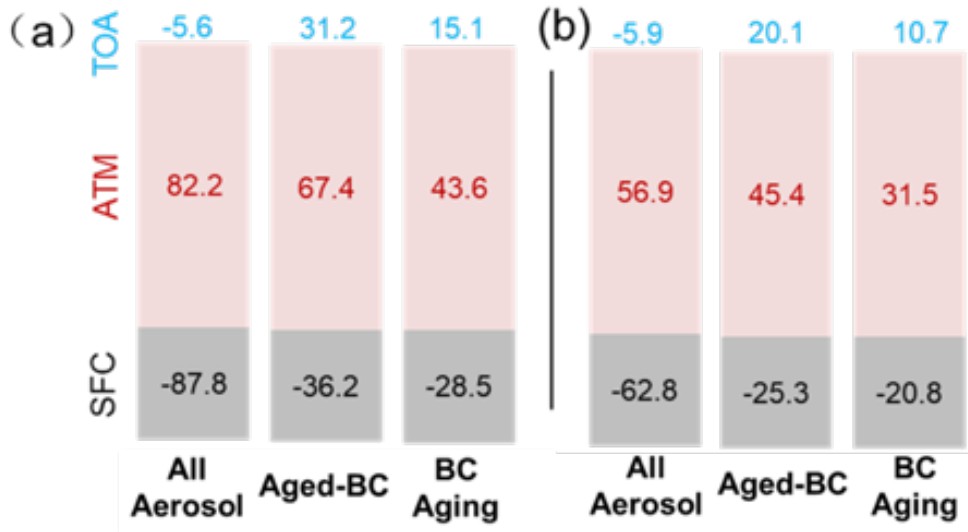

882

**Figure 7.** Aerosol direct radiative forcing for total aerosol (left column), aged-BC (middle column), and BC aging (right column) on the top of the atmosphere (TOA), in the atmosphere (ATM), and at the surface (SFC) for two severe haze days in Beijing. (a) and (b) correspond to EP1 and EP2, respectively. The forcing caused by BC aging corresponds to the difference in the simulations between the fresh-BC and aged-BC cases. The number denotes radiative forcing in the unit of W m$^{-2}$.