# Peer review of "Formation, radiative forcing, and climatic effects of severe regional haze"

_Atmospheric Chemistry and Physics, 2021_

## Author Comment (AC1)

**Reviewer 1** (Reviewer's comments in black and our responses in red)

General comments:

Using in situ measurements, satellite observations and the WRF modeling the authors investigated the contributions of chemical and physical processes to the evolution of haze extremes. Findings show that chemical process that plays a leading role in PM production varies with the development stage of the haze event. And the haze-PBL interactions accelerate the accumulation of aerosol particles and water vapor at the ground level, amplifying the haze severity. The findings help to demonstrate the potential for achieving co-benefits for air quality and climate via black carbon mitigation. I recommend it to be accepted after minor revision. We thank the positive comments to our manuscript by this reviewer and have fully addressed the issues raised by this reviewer below.

Fig. 3 show the concurrent increases in OOA, PM2.5 and Ox concentrations during the transition period. Is that also the case for the clean days? Why or why not? The shaded area for the correlation in Fig. 3 was selected to represent the largest variation in $O_x$, which covered both clean and transition periods. This point has been clarified in the caption.

Generally, the PBLH increases from morning to afternoon, and then decreases to the midnight. While it seems that the rate of growth in PM during the polluted period (from afternoon to midnight) is comparable to that during the transition period (from morning to afternoon) in Figs. 3a-b. Does it mean that photochemistry produced more PM? I suggest the authors to provide additional information on the diurnal evolution of the PBL for the two selected cases, and further explore the role of boundary layer evolution and chemical processes in the development of pollution. We have checked the ceilometer-retrieved PBL heights on 27 September and 4 October, 2013 (see Figure S4) and have provided the following explanations, "Note that both the photochemical production and PBL evolution contribute to PM accumulation at the group level, since the PBL collapse during the daytime leads to vertical dilution for PM. For example, the ceilometer-retrieved PBL height increases about 200 m (150 m) from morning to afternoon and then decreases by about 150 m (400 m) from afternoon to midnight on 27 September (4 October) (Fig. S4). Clearly, the largest PM increase as well as the strong correlation between OOA and $O_x$ during the morning and early afternoon hours indicate that the photochemical PM production dominates the PM increase during the transition period".

In my opinion, the relative humidity calculated by the Nozaki's equation (Lines 402-407 and 501-503) may suffer from large uncertainties that originated from 1). errors in PBL height, especially for the clean cases (Line 257), 2). inappropriate selection of Pasquill stability level (Line 255) due to the lack of cloud fraction measurements. 3) the uncertainty of the empirical formula itself. Since the model simulations agree well with the observations (Fig. s2), why not use the modelled results to show the changes in RH from the ground to the free troposphere? Here we want to estimate how sensitive of RH in response to the change in PBL height, and Tie et al. (2017) has proven that the Nozaki' equation can well serve our purpose. In our calculation, the two inputs of wind and PBL height to Nozaki' equation were based on observations (we used a new observation dataset of PBL height to replace the reanalysis data), and the results show that

our calculated RH is comparable to the observations. We would expect that there are uncertainties by using Nozaki's equation, but we don't need to worry about at least the uncertainty source 2), since there was no cloud effect on Pasquill stability level selection at all because all the days simulated were under cloud-free condition. We took Tie et al. (2017) as a reference when selecting Pasquill stability level in our study.

As for simulation results, the correlation between simulated RH and PBL height was much more complicated since RH calculation in model is influenced by many other factors besides PBL height. It is not straightforward for us to use simulation results to simply quantify the changes in RH in response to the changes in PBL height. As such, the modeling results may not serve our purpose here.

Line 300, the '(Li et al. (2015)' should be '(Li et al., 2015)'.

Corrected as suggested.

I cannot find tables. 1-3 in the manuscript.

The missing tables have been included.

---

## Author Comment (AC2)

**Reviewer 2** (Reviewer's comments in black and our responses in red)

This study investigated the severe regional haze episodes in 2013 over the Northern China Plain through both measurements and simulations. The authors showed strongly correlated OOA and Ox concentrations during transition periods and positive feedback to haze formation by aerosol-PBL interactions during polluted periods. In general, this paper is well written and suitable for publication after the minor comments below are addressed.

We are grateful to the positive comments to our manuscript by this reviewer and have fully addressed the issues raised by this reviewer below.

1. I couldn't find any tables in the main text.
   The missing tables have been included.

2. I couldn't find in the paper how the OOA concentrations were measured or determined. Can the authors add more details of how the OOA data were obtained? Also, can the authors add OOA time series to Figure 1, on top of PM2.5 concentrations?
   Measurements of aerosol chemical compositions were only available in Beijing, which have been previously reported by Guo et al. (2014). We have now stated that "For example, the mass concentrations of various inorganic and organic aerosol species, including oxygenated organic aerosol (OOA), were measured using an aerosol mass spectrometer (AMS) in Beijing (Aiken et al., 2009; Guo et al., 2014)" (Lines 185-188).

3. Line 246: The authors retrieved PBLH from MERRA2, but PBLH can also be estimated by the Nozaki's equation provided here. I wonder if the PBL estimations from the two methods were consistent.
   Recently we obtained a new dataset of ceilometer measurements on PBL heights at Beijing and used these data to replace the MERRA2 to recalculate RH for the two clean days (the PBL height values on hazy days are comparable between the new dataset and the one used in the original manuscript). The PBL height data is presented in Table 3. Using the observed RH as inputs for Nozaki's equation, we can derive the PBL heights as 1289/456 m and 1411/442 m on clean/hazy days for EP1 and EP2, respectively, which are comparable to the observations, i.e., 1180/395 m and 1313/370 m (also see Table 3). Accordingly, we modified Lines 412-415 to "The calculated RH increases from 29% to 68% for EP1 and from 28% to 73% for EP2, when the PBL height decreases from 1180 to 395 m and 1313 to 370 m from clean days to the polluted days for EP1 and EP2, respectively, indicating that the humidity is highly sensitive to the PBL height".

4. Figure 3: Can the authors add R2 between OOA and total PM2.5? It seems that even OOA only accounted for less than 1/3 of PM2.5 mass, their concentrations were highly correlated. In this way, would PM2.5 also show a good correlation with Ox? In addition, it would be helpful to add the same set of graphs for clean days.
   The PM$_{2.5}$ mass consists of both primary and secondary components, while the OOA corresponds to only the secondary component. Since O$_x$ represents the level of oxidants relevant only to secondary aerosol, a simple correlation between PM$_{2.5}$ and Ox would be less meaningful.

The shaded area for the correlation in Fig. 3 was selected to represent the largest variation in $O_x$, which covered both clean and transition periods. This point has been clarified in the caption.

5. Figure 4: Can the authors add the PBLH profile for transition days?
   We added the diurnal cycle of PBL height of transition days (30 μg m$^{-3}$ < daily mean PM$_{2.5}$ < 200 μg m$^{-3}$) to Fig. 4b.

6. Line 405: How did these calculated RH compare to the measured RH in Figure 1?
   We have described the measured RH changes in the original manuscript, and quoted here for reference: "the measured RH increases greatly during the two episodes (Fig. 1d), i.e., from about 18%-19% on the clean days (25 September and 2 October) to 53%-55% on the polluted days (28 September and 5 October)" (Lines 412-415). The calculated RH was reported in Table 3, showing that RH is slightly overestimated in all the four days relative to observations.

7. Line 406: where did the 1000 m decrease come from? I only find in Line 371 "the simulated maximal height of PBL under the polluted condition is reduced by more than 300 m relative to the clean condition".
   In the previous manuscript we extracted the PBL height from MERRA2 because we didn't get the available observations on the two clean days, and the difference between the clean and hazy days are about 1000 m. Recently we obtained a new dataset of PBL height based on ceilometer measurements (see Table 3) for the two clean days and we used it to replace MERRA2 data.
   The statement "the simulated maximal height of PBL under the polluted condition is reduced by more than 300 m relative to the clean condition" is based on simulation results.

8. Figure 4&5: Both figures show PBLH diurnal profiles, but with very different magnitude. Only looking at clean days, Figure 5 PBLH is only half of that in Figure 4. Why are the simulated results so different from the MERRA2 reanalysis?
   There are several possible factors which may contribute the relatively large discrepancy in PBL heights between Fig. 4 and 5: 1) Fig. 4 shows the statistics of PBL heights under all the clean and hazy days in 2013 based on MERRA2 data, with error bars denoting uncertainties. As such, the PBL height values shown here include both warm and cold seasons. As we know, the PBL during warm season (e.g., summer) is much higher than that during cold season because of the much higher temperature during warm season than cold season. Fig. 5 shows the PBL heights in late fall when the temperatures were already low, so the lower PBL heights during these simulation days are expected; 2) MERRA2 has much coarser spatial resolution (i.e., 0.5° by 0.625°) than the simulations in this study (i.e., 2 km by 2 km), which might cause the discrepancy between the two.

9. Figure 5: Can the authors either add legend or rewrite the caption? It should be clearly stated that the top 6 figures and bottom 6 figures are for two episodes.
   In the caption (line 894), we have indicated that "(a-f) are for EP1 and (g-n) are for EP2".

---

## Author Response (AR2)

**March 20, 2022**

**Reviewer 1 (Reviewer's comments in black and our responses in red)**

General comments:

The study conducted by Lin et al. shed some new light on the mechanism for the occurrence of the haze extreme. In the revised manuscript, the authors have added some critical information, such as the missing tables and the new observation dataset of PBL height, to further support their conclusions. And most of concerns raised by the referees have been addressed. However, there are still some language errors in the current version and the authors should examine the text more carefully to avoid these problems.

We are grateful to the positive comments to our manuscript by this reviewer and have fully addressed the issues raised by this reviewer below.

For example, Line 347, 'at the group level' should be 'at the ground level'. Done.

Lines 347-348, the collapse of atmospheric boundary layer will lead to pollution accumulation rather than the vertical dilution for PM. We modified the statement as "Note that both the photochemical production and PBL evolution contribute to PM accumulation at the ground level, since the PBL development during the daytime leads to vertical dilution" (Lines 346-348).

We also found other unclear statements/typos, which were modified/corrected as below:

Lines 258-261: The statement was modified to "The input of the PBL height for the both clean and aged-BC cases were based on ceilometer measurements, showing that the PBL height increases by about 700~900 m from the aged-BC cases to the clean cases".

Line 475: Section no. "3.5" should be "3.4".

What's more, it is important to include some discussion to strengthen the reliability of the model simulation results in the manuscript. Here are some observational evidences for the impact of aerosol-PBL interaction on near-surface haze pollution on different time scales.
1. Dong et al., 2017. Opposite long-term trends in aerosols between low and high altitudes: a testimony to the aerosol–PBL feedback, Atmos. Chem. Phys., 17, 7997–8009.
2. Huang et al., 2018. Impact of Aerosol-PBL Interaction on Haze Pollution: Multiyear Observational Evidences in North China. Geophysical Research Letters, 45, 8596–8603.
3. Su et al., 2020. The significant impact of aerosol vertical structure on lower atmosphere stability and its critical role in aerosol–planetary boundary layer (PBL) interactions. Atmos. Chem. Phys., 20, 3713–3724.

We have added some discussions about the reliability of the model simulation results in the

manuscript as below:

Lines 387-390: The extent of warming in the upper boundary layer and cooling at the surface due to the aerosol effect simulated at Beijing in this study are consistent with the observational analysis on North China by Huang et al. (2018), suggesting that our simulations can well reproduce the aerosol radiative effect under severe regional haze condition.

Lines 411-418: The significant aerosol-PBL interaction and its impact on surface air pollution revealed in our simulation sensitivity studies are also evident in multiple observation-based studies in China (Dong et al., 2017; Huang et al., 2018; Su et al., 2020). However, there might exist certain uncertainties in evaluating the aerosol impacts on PBL development based our simulation experiments as previous observational analysis like Dong et al. (2017) and Su et al. (2020) has pointed out that the aerosol-PBL interaction also varies with the aerosol vertical structure but an exponential decreasing aerosol profile was assumed and fixed in our simulations.

Lines 470-473: The significant role of BC in atmospheric heating is also evident in long-term observations, e.g., Huang et al. (2018) has also proven that the heating in the atmosphere was mainly caused by absorbing aerosols like BC.